# Postoperative Analgesic Effectiveness of Peripheral Nerve Blocks in Cesarean Delivery: A Systematic Review and Network Meta-Analysis

**DOI:** 10.3390/jpm12040634

**Published:** 2022-04-14

**Authors:** Choongun Ryu, Geun Joo Choi, Yong Hun Jung, Chong Wha Baek, Choon Kyu Cho, Hyun Kang

**Affiliations:** 1Department of Anesthesiology and Pain Medicine, College of Medicine, Chung-Ang University, Seoul 06911, Korea; sadmime@naver.com (C.R.); pistis23@naver.com (G.J.C.); jyh623@cau.ac.kr (Y.H.J.); nbjhwa@naver.com (C.W.B.); 2Department of Anesthesiology and Pain Medicine, College of Medicine, Konyang University, Daejeon 35365, Korea; kl648227@naver.com

**Keywords:** cesarean section, nerve block, network meta-analysis, obstetrical analgesia pain, systematic review

## Abstract

The purpose of this systematic review and network meta-analysis was to determine the analgesic effectiveness of peripheral nerve blocks (PNBs), including each anatomical approach, with or without intrathecal morphine (ITMP) in cesarean delivery (CD). All relevant randomized controlled trials comparing the analgesic effectiveness of PNBs with or without ITMP after CD until July 2021. The two co-primary outcomes were designated as (1) pain at rest 6 h after surgery and (2) postoperative cumulative 24-h morphine equivalent consumption. Secondary outcomes were the time to first analgesic request, pain at rest 24 h, and dynamic pain 6 and 24 h after surgery. Seventy-six studies (6278 women) were analyzed. The combined ilioinguinal nerve and anterior transversus abdominis plane (II-aTAP) block in conjunction with ITMP had the highest SUCRA (surface under the cumulative ranking curve) values for postoperative rest pain at 6 h (88.4%) and 24-h morphine consumption (99.4%). Additionally, ITMP, ilioinguinal-iliohypogastric nerve block in conjunction with ITMP, lateral TAP block, and wound infiltration (WI) or continuous infusion (WC) below the fascia also showed a significant reduction in two co-primary outcomes. Only the II-aTAP block had a statistically significant additional analgesic effect compared to ITMP alone on rest pain at 6 h after surgery (−7.60 (−12.49, −2.70)). In conclusion, combined II-aTAP block in conjunction with ITMP is the most effective post-cesarean analgesic strategy with lower rest pain at 6 h and cumulative 24-h morphine consumption. Using the six described analgesic strategies for postoperative pain management after CD is considered reasonable. Lateral TAP block, WI, and WC below the fascia may be useful alternatives in patients with a history of sensitivity or severe adverse effects to opioids or when the CD is conducted under general anesthesia.

## 1. Introduction

Cesarean sections cause moderate-to-severe acute postoperative pain. Furthermore, more than 10% of mothers may experience persistent post-cesarean delivery (CD) pain, which may persist for more than 3–6 months after surgery [1,2]. Inadequately controlled postoperative pain may delay the mother’s recovery, disturb breastfeeding, and interfere with maternal-neonatal bonding [3]. Therefore, optimal postoperative analgesia is a major concern for both parturients and obstetric anesthesiologists.

Low-dose intrathecal morphine (ITMP) is considered the gold standard for postoperative analgesia after a cesarean section under spinal anesthesia [4]. However, it has the potential for unwanted opioid-related adverse effects, such as pruritus, nausea, vomiting, urinary retention, and sedation, which could increase the requirement for additional medications.

Currently, multimodal analgesic strategies are widely used after cesarean section to reduce opioid consumption and provide synergistic or additive analgesia [5]. Several peripheral nerve block (PNB) techniques have been investigated to assess their potential analgesic benefit for post-CD pain: erector spinae plane (ESP) block, quadratus lumborum (QL) block, ilioinguinal-iliohypogastric (II-IH) nerve block, transversus abdominis plane (TAP) block, rectus sheath (RS) block, and surgical wound infiltration (WI) [6]. To date, more than 10 published meta-analyses have compared the analgesic effect of these abdominal wall fascial plane blocks. However, most of them concluded that there was no further improvement in analgesic outcomes when intrathecal morphine was combined [7,8,9,10]. Conversely, there are several different approaches for each block technique, and the spread of the local anesthetic may vary depending on the anatomical location of each approach. Resultantly, analgesic outcomes may differ depending on the approach [6]. Consequently, previously published meta-analyses that were not classified according to each approach may have some limitations in the interpretability of their results and may provide incorrect information to clinicians.

This systematic review and network meta-analysis (NMA) aimed to compare the analgesic effectiveness of all types of PNBs to decrease postoperative pain and opioid consumption after cesarean section. Specifically, in this study, we analyzed the results by considering each anatomical approach of the PNB method and whether intrathecal morphine was used in combination during spinal anesthesia as an individual comparator. Ultimately, we sought to answer the following conclusions: (1) Is ITMP still sufficient as the gold standard, (2) which method is the most effective strategy for post-CD analgesia, and (3) which PNB methods have additional analgesic effects when combined with ITMP?

## 2. Materials and Methods

This systematic review and NMA on multimodal post-CD pain was conducted in accordance with the protocol recommended by the Cochrane Collaboration [11] and reported according to the PRISMA extension for NMA guidelines [12]. We used a pre-designed protocol to review and specifically evaluate the results of randomized controlled trials (RCTs) that compared the postoperative analgesic outcomes of all types of PNB methods and ITMP after CD. The protocol was registered with the PROSPERO network (registration number: CRD42021264372; https://www.crd.york.ac.uk/prospero, accessed on 1 December 2021).

### 2.1. Eligibility Criteria

We included full reports of RCTs that investigated the postoperative analgesic effects of all types of PNB methods and intrathecal or epidural morphine in CD. A detailed description of the inclusion and exclusion criteria is provided in the Appendix A.

### 2.2. Search Strategy and Study Selection

We searched MEDLINE, EMBASE, the Cochrane Central Register of Controlled Trials (CENTRAL), and Google Scholar using search terms related to PNB and ITMP for the management of post-CD pain from inception until July 2021. Two investigators (C.W.B. and C.K.C.) independently selected the studies.

A detailed description of the literature search strategy and study selection is provided in Appendix A.

### 2.3. Data Extraction and Management

Using standardized extraction, the following data were extracted by two independent investigators (C.R. and H.K.): (1) title; (2) name of the first author; (3) name of the journal; (4) year of publication; (5) country; (6) language; (7) primary anesthesia details and regimen (general versus. neuraxial); (8) block technique and approach used; (9) number of subjects; (10) the type and dose of drug used; (11) nature of primary and secondary outcomes investigated; (12) supplemental postoperative analgesia regimen. A detailed description of the data extraction and data management is provided in Appendix A.

### 2.4. Quality Assessment

The quality of all included studies was independently assessed by two investigators (C.R. and H.K.), using version 2 of the Cochrane RoB tool for randomized trials. RoB 2 was evaluated by considering the following five potential sources of bias: (1) bias arising from the randomization process, (2) bias due to deviations from intended interventions, (3) bias due to missing outcome data, (4) bias in outcome measurements, and (5) bias in the selection of the reported results. We judged all five domains in each article according to a series of questions, “signaling questions,” presented in the Cochrane RoB 2 to elicit information about features of the trial that are relevant to the risk of bias.

Thereafter, we evaluated the overall risk of biased judgment according to domain-level judgments. The methodology for each domain was assigned as “high risk of bias,” “some concerns,” or “low risk of bias” to reflect the risk of bias [13].

### 2.5. Quality of the Evidence

Evidence grade was determined using the Grading of Recommendations Assessment, Development, and Evaluation (GRADE) system, which uses a sequential assessment of evidence quality, followed by an assessment of the risk-benefit balance and a subsequent judgment on the strength of the recommendations.

### 2.6. Statistical Analysis

A frequentist NMA was performed using STATA software (version 15; StataCorp LP, College Station, TX, USA) based on mvmeta with NMA graphical tools developed by Chaimani et al. [14]. Additionally, to test the robustness of the results of the frequentist random NMA, we also conducted Bayesian NMA using fixed and random effects models and with Markov Chain Monte Carlo methods using the R statistical package gemtc [15]. We evaluated the similarity, transitivity, and consistency assumptions for the NMA. We also performed a network meta-regression analysis to test the possible causes of heterogeneity. A detailed description of the statistical analysis is provided in Appendix A.

## 3. Results

### 3.1. Study Selection

We initially identified 1164 unique citations from MEDLINE, EMBASE, the Cochrane Central Register of Controlled Trials, and Google Scholar database. Additionally, we retrieved 10 more articles from the reference lists of related meta-analyses. A PRISMA flow chart of the study selection is shown in. After the removal of duplicates (129 studies), we conducted extensive screening of the individual titles and abstracts of 1045 studies. In the first stage of study selection, the kappa value between the two investigators was 0.768. A total of 98 studies that met the predefined definitions of PICO-SD (population, intervention, comparator, outcome and study design) remained, whose eligibility we evaluated via full-text reviews. Among these, 22 additional studies were excluded for the reasons described in Figure 1 and Appendix A. Finally, we included 76 RCTs in this systematic review and the NMA [16,17,18,19,20,21,22,23,24,25,26,27,28,29,30,31,32,33,34,35,36,37,38,39,40,41,42,43,44,45,46,47,48,49,50,51,52,53,54,55,56,57,58,59,60,61,62,63,64,65,66,67,68,69,70,71,72,73,74,75,76,77,78,79,80,81,82,83,84,85,86,87,88,89,90,91]. In the second stage of study selection, the kappa value between the two investigators was 0.939.

### 3.2. Study Characteristics

The characteristics of included studies are presented in Table 1. In papers published since 1991, we identified seven types of PNB methods: ESP block, transverse fascial plane (TFP) block, QL block, II-IH nerve block, TAP block, RS block, and surgical WI. Although there are several specific approach techniques for each block, only the following approach techniques were included in this study through data extraction: QL block (anterior, aQL; posterior, pQL, combined anterior and posterior; apQL, and lateral approach; lQL), TAP block (anterior; aTAP, lateral; lTAP, combined subcostal and lateral; slTAP, posterior approach; pTAP), continuous wound infusion (catheter insertion above or below the fascia; WC_above or WC_below, respectively). Consequently, we identified 28 different postoperative analgesia strategies (164 directly compared group), except for the non-active control group (no intervention), either alone or in various combinations. Except for non-active controls (45 studies, 27.4%, 1835 patients), ITMP (active control, 24 studies, 14.6%, 891 patients), lTAP block (20 studies, 12.2%, 650 patients), and WI, 15 studies, 9.1%, 662 patients) were compared the most. A detailed description is provided in Appendix A.

### 3.3. Study Quality Assessment

The risk of bias assessment in the included studies using the Cochrane RoB 2 is presented in Appendix A). When judging the overall risk of bias, only 13 studies had a low risk of bias in all domains. A detailed description of the study quality assessment is provided in Appendix A.

### 3.4. Synthesis of Results

#### 3.4.1. Primary Outcomes

##### Pain at Rest 6 h after Surgery

Although a total of 60 studies (4672 patients) measured the pain at rest 6 h after surgery, as one study was separated from the loops [72], we performed the NMA excluding that study. Therefore, 59 studies (4622 patients) were analyzed.

The network plot of all eligible comparisons for this endpoint is shown in Figure 2. Although all 25 analgesic management strategies (nodes) were connected to the network, two comparisons (control and lTAP block) were more directly comparable than the other 23 nodes. There was no evidence of network inconsistency (χ^2^ (14) = 17.29, *p* = 0.241). There were 19 triangular loops, and two quadratic loops closed in the network from the comparison of pain at rest 6 h after surgery. Triangular loops are formed by three interventions all compared with each other, and quadratic loops are formed by four interventions, each of which is compared exactly with two other interventions in the loop.

Five loops (aQL/pQL/apQL [50], aQL/pQL/EDMP [50], aQL/apQL/EDMP [50], pQL/apQL/EDMP [50], and RS/ITMP + RS/ITMP [60]) were formed only by multi-arm trials. Although almost all loops showed no significance in the local inconsistency between the direct and indirect point estimates, five loops (Control/aQL/WC_below/EDMP, Control/pQL/WC_below/EDMP, pQL/ITMP + pQL/ITMP, Control/pQL/lTMP + pQL, and Control/aTAP/II-IH) showed significant inconsistency (Appendix A).

ITMP in conjunction with the ilioinguinal nerve and aTAP block (ITMP + II-aTAP) showed a lower pain at rest 6 h after surgery than the non-active control group (no intervention) in terms of 95% CI (−14.87, −4.69) and PrI (−16.41, −3.15). Conversely, ITMP + II-IH, ITMP + lTAP, ITMP, WC_below, lTAP, and WI showed lower pain than the controls only in terms of the 95% CI (Figure 3). Insignificance in the 95% PrIs suggests that any future RCTs could change the significance of the efficacy of these comparisons.

The rankogram shows the distribution of the estimated probability for each intervention to achieve each of the possible ranks. Appendix A shows that ITMP + II-aTAP followed by ITMP + II-IH, ITMP + lTAP and ITMP + RS had the lowest pain at rest 6 h after surgery. A cumulative ranking plot was drawn, and the SUCRA probabilities of the different interventions for this outcome were calculated (Appendix A). The expected mean rankings and SUCRA values for each intervention are presented in Figure 4. According to the SUCRA value, the pain at rest 6 h after surgery was lower in the order of ITMP + II-aTAP (99.4%), followed by ITMP + II-IH (88.9%), ITMP + lTAP (77.8%), and ITMP + RS (77.0%). The SUCRA values from the Bayesian model are like those from the frequentist model, which demonstrates the robustness of our analysis. The comparison-adjusted funnel plots showed that the funnel plots were symmetrical around the zero line, which suggested a less likely publication bias (Appendix A).

The network diagnostics using trace and density plots showed that the model convergence was valid in both models (Appendix A). However, the Gelman–Rubin–Brooks method with a potential scale reduction factor (PSRF) and DIC showed that the random effect model was a better fit for the data (Appendix A; Appendix A). Thus, we analyzed the data using a random effect model.

Forest plots, node splitting plots, and rankograms generated by the Bayesian model were also generated (Appendix A), which shows results like those of the frequentist model.

##### Postoperative Cumulative 24 h Morphine Equivalent Consumption (mg)

A total of 44 studies (3360 patients) presented postoperative cumulative 24 h morphine equivalent consumption. In most studies, opioids were administered using an intravenous patient-controlled analgesia (PCA) device, and in some cases, intermittent IV bolus injection was performed according to the patient’s pain scale. In all studies in which IV morphine was administered through a PCA device after the surgery, basal infusion was not performed, but a bolus dose and lockout time were set up.

The network plot of all eligible comparisons for this endpoint is shown in Figure 2B. Although all 21 analgesic management strategies (nodes) were connected to the network, three comparisons (Control, ITMP, and lTAP) were more directly comparable to the other 18 nodes. There was no evidence of network inconsistency (χ^2^ (7) = 7.59, *p* = 0.371). There were 14 triangular loops closed in the network based on the comparison of this outcome.

Five loops (aQL/pQL/apQL [50], aQL/pQL/EDMP [50], aQL/apQL/EDMP [50], pQL/apQL/EDMP [50], and RS/ITMP + RS/ITMP [60]) were formed only by multi-arm trials. There was no evidence of local inconsistency between the direct and indirect point estimates (Appendix A).

ITMP + II-aTAP showed a lower morphine consumption at 24 h than the control in terms of 95% CI (−44.47, −10.06) and PrI (−52.70, −1.82) at the same time. LQL, ITMP + RS, ITMP + WI, ITMP + IPLA (intraperitoneal local anesthetic instillation), ITMP, ITMP + II-IH, lTAP, WI, TFP, and WC_below showed a lower morphine consumption than the control only in terms of the 95% CI (Figure 3B).

The rankograms showed that ITMP + II-aTAP followed by lQL and ITMP + RS had the lowest postoperative cumulative 24 h morphine equivalent consumption (Appendix A). A cumulative ranking plot was drawn, and the SUCRA probabilities of the different interventions for this endpoint were calculated (Appendix A). The expected mean rankings and SUCRA values for each intervention are presented in Figure 4B. According to the SUCRA value, the postoperative cumulative 24 h morphine equivalent consumption was lower in the ITMP + II-aTAP (88.4%), followed by lQL (75.0%), ITMP + RS (71.3%), and ITMP + WI (69.4%). The comparison-adjusted funnel plots showed that the funnel plots were symmetrical around the zero line, which suggested a less likely publication bias (Appendix A).

The network diagnostics using trace and density plots showed that model convergence was valid in both models (Appendix A). However, the Gelman-Rubin-Brooks methods with PSRF and DIC showed that the random effect model was a better fit for the data (Appendix A; Appendix A). Thus, we analyzed the data using a random effect model.

Forest plot, node splitting plot, rankogram, and SUCRA values from the Bayesian model showed similar results to those from the frequentist model, which showed the robustness of our analysis (Appendix A; Figure 4B).

#### 3.4.2. Secondary Outcomes

##### Pain at Rest 24 h after Surgery

The network plot of all eligible comparisons for this outcome is shown in Figure 2C. Continuous wound infusion below the fascia (WC_below) showed lower pain than the control only in terms of the 95% confidence interval (Figure 3C). According to the SUCRA value, the pain at rest 24 h after surgery was lower in the order of the slTAP (83.6%), followed by ITMP + II-aTAP (82.1%), IPLA (80.1%), and WC_below (79.8%) (Appendix A; Figure 4C). Inconsistency and publication bias were checked (Appendix A). The network diagnostics showed that the random effect model was a better fit for the data (Appendix A; Appendix A). Forest plot, node splitting plot, rankogram, and SUCRA values from the Bayesian model showed similar results to those from the frequentist model, which showed the robustness of our analysis (Appendix A; Figure 4C).

##### Dynamic Pain 6 h after Surgery

The network plot of all eligible comparisons for this endpoint is shown in Figure 2D. WI showed lower pain than the control only in terms of 95% CI (Figure 3D). According to the SUCRA value, the dynamic pain at 6 h after surgery was lower in the order of WI (78.9%), followed by ESP (72.4%), slTAP (71.7%), and apQL (64.4%) (Appendix A, and Figure 4D). The comparison-adjusted funnel plots showed that the funnel plots were symmetrical around the zero line, which suggested a less likely publication bias (Appendix A). Inconsistency and publication bias were checked (Appendix A).

The network diagnostics showed that the random effect model was a better fit for the data (Appendix A; Appendix A). Forest plot, node splitting plot, rankogram, and SUCRA values from the Bayesian model showed similar results to those from the frequentist model, which showed the robustness of our analysis (Appendix A and Figure 4D).

##### Dynamic Pain 24 h after Surgery

The network plot of all eligible comparisons for this endpoint is shown in Figure 2E. EDMP, apQL, WC_below, and lTAP showed lower pain than the control in terms of the 95% CI only (Figure 3E). According to the SUCRA value, the dynamic pain at 24 h after surgery was lower in the order of EDMP (96.7%), followed by apQL (89.2%), ESP (74.2%), and WC_below (71.3%) (Appendix A, and Figure 4E). Inconsistency and publication bias were checked (Appendix A).

The network diagnostics showed that the random effect model was a better fit for the data (Appendix A; Appendix A). Forest plot, node splitting plot, rankogram, and SUCRA values from the Bayesian model showed similar results to those from the frequentist model, which showed the robustness of our analysis (Appendix A and Figure 4E).

##### Time to First Analgesic Request (h)

The network plot of all eligible comparisons for this endpoint is shown in Figure 2F. ESP showed a longer time to first analgesic request than controls in terms of the 95% CI (30.84, 42.04) and PrI (19.95, 52.94) at the same time. Additionally, ITMP, pTAP, II-IH, lTAP, ITMP + WI, TFP, and pQL showed a longer time to first analgesic request than controls only in terms of the 95% CI (Figure 3F). According to the SUCRA value, the time to first analgesic request was longer in the order of the ESP (100.0%), followed by pQL (91.2%), TFP (87.5%), ITMP + WI (63.8%), and lTAP (62.9%) (Appendix A and Figure 4F). Inconsistency and publication bias were checked (Appendix A).

The network diagnostics showed that the random effect model was a better fit for the data (Appendix A; Appendix A). The forest plot, node splitting plot, rankogram, and SUCRA values from the Bayesian model showed similar results to those from the frequentist model, which showed the robustness of our analysis (Appendix A and Figure 4F).

Detailed descriptions of the results of secondary outcomes are provided in Appendix A.

### 3.5. Quality of the Evidence

Six outcomes were evaluated using the GRADE system. The evidence quality was moderate for pain at rest 6 h after surgery, postoperative cumulative 24 h morphine equivalent consumption, and the time to first analgesic request, and low for pain at rest 24 h after surgery, dynamic pain at 6 h after surgery, and dynamic pain at 24 h after surgery. A detailed description of the quality of the evidence assessment is provided in Appendix A and Table 2.

## 4. Discussion

Our systematic review and NMA showed the potential analgesic role of combined II-aTAP block in conjunction with ITMP in providing post-cesarean analgesia. It is the most effective analgesic strategy with lower rest pain at 6 h and morphine consumption at 24 h after surgery compared to the non-active control group (no intervention) in terms of the 95% CI and PrI at the same time. Additionally, ITMP, ITMP+ II-IH, lTAP, WI, and WC_below showed a significant reduction in both co-primary outcomes compared to the non-active control in terms of the 95% CI only. On the other hand, in terms of the additional effect of using PNB with ITMP, only II-aTAP had a statistically significant additional effect compared to ITMP alone on pain at rest 6 h after surgery (mean difference (95% CI) (−7.60 (−12.49, −2.70)) (Appendix A).

NMA is a useful tool for comparative effectiveness research. We compared the analgesic effects of all kinds of post-cesarean analgesic strategies extracted from all included RCTs; therefore, through this NMA, we will be able to explore the answers to the following questions.

First, is ITMP monotherapy still sufficient as the gold standard for post-cesarean analgesia? It is known to provide long-lasting analgesic effects up to 14–36 h [92]. Regarding the duration of action, a meta-analysis designed to determine the analgesic effect according to the difference in ITMP dose after CD concluded that higher doses (>100 μg) of ITMP prolonged analgesia compared with lower doses (50–100 μg) [93]. The range of mean times to first analgesic request was 13.8 h to 39.5 h and 9.7 h to 26.6 h in the higher and lower doses groups, respectively. In our study, a total of 24 included studies (*n* = 891, 133.8 ± 56.1 μg) administered ITMP, of which 15 used lower doses, and nine used higher doses of morphine. Despite the use of relatively higher doses of morphine, ITMP showed no statistically significant differences compared with non-active controls in the three secondary outcomes. It prolongs the time to first analgesic request (mean difference (95% CI); 3.96 (1.20, 6.73)), reduces cumulative 24-h morphine consumption (−16.55 (−23.89, −9.21)), and relieves rest pain at 6 h after surgery (−2.18 (−3.59, −0.77)) compared to the non-active control. Although none of the PNB was superior to ITMP in all the three outcomes described above, when PNBs were used together with ITMP, many strategies had greater mean differences and SUCRA values than ITMP monotherapy. Additionally, considering the mean difference in the time to first analgesic request (3.96 h), the analgesic duration of ITMP is thought to be shorter than the result derived from a previous meta-analysis, which was analyzed in fewer studies than ours [93]. Consequently, although ITMP is still simple and effective, it is not the best method in terms of SUCRA values, and it cannot be an option for general anesthesia.

Then, what is the most effective post-cesarean analgesic strategy? PNBs are increasingly used as an important component of postoperative multimodal analgesia, and many RCTs and meta-analyses have been conducted to compare their post-cesarean analgesic effect. As described in the Appendix A, the analgesic outcomes of PNBs may differ depending on the anatomical approach, and the results that do not distinguish each approach are likely to provide misinformation to clinicians. Therefore, we designed each approach technique as an individual comparator for a more accurate comparison of the post-cesarean analgesic effects of PNBs.

Based on the SUCRA value, ITMP + II-aTAP was ranked as the most effective strategy, with the lowest 24-h morphine consumption (88.4%) and the lowest rest pain at 6 h (99.4%). It also had a statistically significant mean difference compared with non-active controls in terms of the 95% CI and PrI. In particular, it significantly reduced rest pain at 6 h compared to all types of included analgesic strategies, except one (Appendix A). Additionally, it has an opioid-sparing effect (mean difference in 24-h morphine consumption: 27.26 mg, Figure 3B). Reducing the use of opioids improves postoperative recovery and minimizes the risk of opioid-related adverse events. A description of why this combined PNB was most effective is provided in the Appendix A.

Additionally, considering analgesic duration, ESP block was most effective in prolonging the time to the first analgesic request in terms of both 95% CI and PrI. The SUCRA value was determined to be 100%. The ESP block is a relatively new paraspinal regional anesthesia technique described for the first time in 2016 [94]. It aims to disperse local anesthetics between the transverse process of the vertebra and erector spinae muscle and then spread it into the paravertebral space of 3 or 4 vertebral levels cranially and caudally [62]. Therefore, it can offer both somatic and visceral analgesia.

Most PNBs, including ITMP, were ineffective in late postoperative pain at 24 h in our study (Figure 3C,E). Only WC_below (−1.60 (−2.48, −0.72), −1.64 (−2.81, −0.47)) and lTAP (−0.61 (−1.18, −0.03), −0.85 (−1.66, −0.05)) showed statistically significant lower pain scores in both rest and dynamic pain at 24 h, respectively (Figure 3C,E). WC_below was much more effective than lTAP in terms of the mean difference and SUCRA value. Therefore, for late post-cesarean pain after 24 h, continuous wound infusion of local anesthetics via a catheter below the fascia is recommended.

Third, which PNB methods have an additional analgesic effect when used with ITMP? In our study, only II-aTAP had a statistically significant additional analgesic effect in pain at rest 6 h after surgery (−7.60 (−12.49, −2.70), Appendix A). In any other included intervention, there was no statistically significant difference in terms of the 95% CI.

Finally, through this study, in addition to the predefined questions, we obtained the following additional information.

First, synthetically, six analgesic strategies are written in the first paragraph of the Discussion section that significantly reduced both the rest pain at 6 h after surgery and 24-h morphine consumption at the same time. Therefore, it is reasonable to use these strategies for post-cesarean analgesia. In addition, lTAP, WI, and WC_below showed favorable results, even in the absence of ITMP (Figure 3A,B). Therefore, these three PNB techniques may be useful alternatives if the patient reports a history of sensitivity or severe adverse effects to opioids or when the CD is conducted under general anesthesia.

Second, among the articles finally included in this meta-analysis, the PNB method with the most comparisons was lTAP (20 studies, 650 patients). Abdallah et al. [95] reported that pTAP appeared to lower not only rest and dynamic pain but also morphine consumption for up to 48 h compared to the controls in their meta-analysis. On the contrary, lTAP showed notably more favorable results in our study, except for dynamic pain at 6 h after surgery, compared with pTAP. We believe that the results have changed because 21 articles published after this meta-analysis were retrieved and analyzed in our study.

The QL block is known to have additional effects on visceral pain compared to TAP, which is expected to show better results, but not in our study. LQL was ranked as the second most effective strategy to reduce 24-h morphine consumption (SUCRA value 75.0%), apQL was the second most effective strategy to lower dynamic pain at 24 h after surgery (89.2%), and pQL was also the second most effective strategy to prolong the time to first analgesic request (91.2%) compared to non-active controls in the absence of ITMP. However, there were no statistically significant differences with any approach to the TAP block in terms of mean difference and 95% CI.

Our study has several limitations. First, as with all meta-analyses, there were inevitable heterogeneities, especially in methodologies among the finally included 76 RCTs, which may be attributed to variability in the doses of short-acting opioids such as fentanyl or sufentanil as adjuvants for spinal anesthesia, doses of intrathecal or epidural morphine, doses of local anesthetics for spinal anesthesia and PNBs as described in the results section, and a wide variety of combinations of analgesics used for postoperative multimodal analgesic regimens such as opioids, acetaminophen, and non-steroidal anti-inflammatory drugs. Second, as most of the trials were conducted in a single center using a small scale, there was a possibility of a lack of evidence on some interventions. Third, although II-aTAP in conjunction with ITMP was the most effective postoperative analgesic strategy after CD, it was compared in only one trial with a small sample size (*n* = 50 patients); therefore, there is a risk of overestimation of effect size. Fourth, many of the included trials had a risk of bias, as described in the results section. Specifically, in performance bias, the studies that compared different PNB methods were unable to blind the investigator who performed the intervention. However, all PNBs were performed in the operating room or post-anesthesia care unit (PACU), and all our primary and secondary outcomes were measured after discharge from the PACU. Additionally, among those studies, there was no case in which the domain of ‘bias in the measurement of the outcome’ was judged as high risk at the same time.

Despite these limitations, the current NMA has several strengths. First, this is the first NMA to compare and quantify the rank order of the relative effect of various strategies for post-CD pain, which may help patients, anesthesiologists, obstetricians, and policy makers in making evidence-based decisions. Second, a rigorous methodology based on a pre-planned protocol was applied. Specifically, we performed both frequentist and Bayesian NMA to test the robustness of the results and network meta-regression analysis to test the possible cause of heterogeneity.

## 5. Conclusions

This NMA shows that ITMP monotherapy is still simple and effective for post-cesarean analgesia. However, it is not the best method in terms of SUCRA values, and it cannot be an option for general anesthesia. Combined II-aTAP block in conjunction with ITMP for spinal anesthesia is the most effective analgesic strategy with lower rest pain at 6 h and morphine consumption at 24 h after surgery compared with non-active controls. Additionally, ITMP, II-IH nerve block in conjunction with ITMP, lTAP block, and WI or WC_below showed a significant reduction in the two co-primary outcomes as well. Therefore, it is reasonable to use PNBs for postoperative pain management after CD. In particular, lTAP block, WI, and WC_below may be useful alternatives if the patient reports a history of sensitivity or severe adverse effects to opioids or when the CD is conducted under general anesthesia. Finally, only the II-aTAP block had a statistically significant additional effect compared to ITMP alone on rest pain at 6 h after surgery.

## Figures and Tables

**Figure 1 jpm-12-00634-f001:**
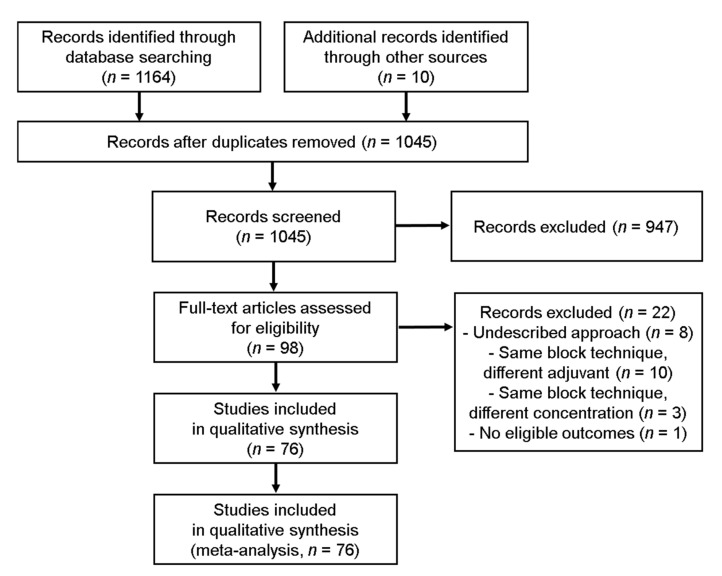
PRISMA flow diagram of literature search and selection.

**Figure 2 jpm-12-00634-f002:**
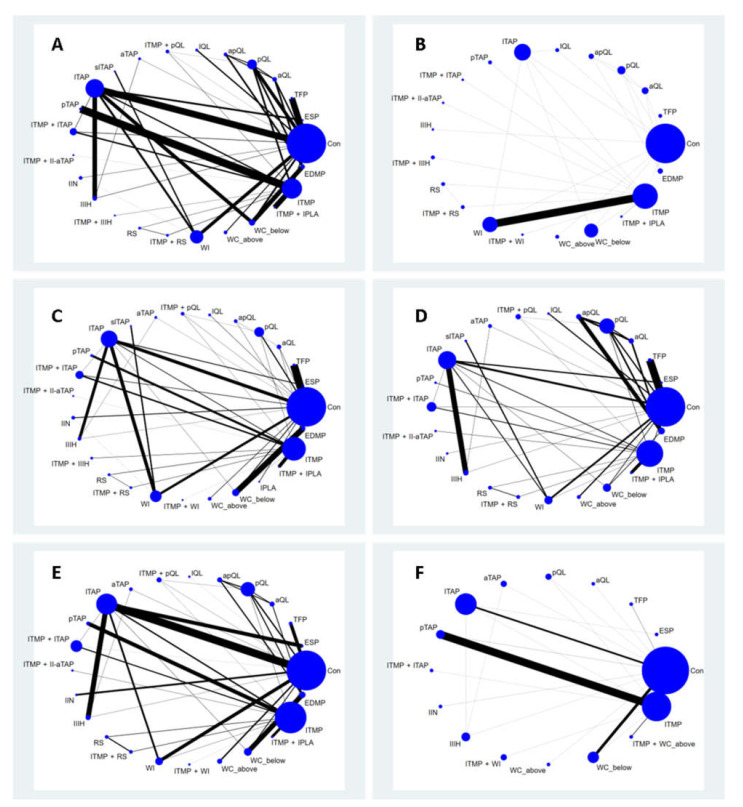
Network plots of direct comparisons for all included studies for outcomes. Each postoperative analgesia strategy is depicted by a node that is weighed based on the number of subjects who were randomized to that intervention. Edges between the nodes show the eligible direct comparisons among those interventions, and their width is weighed based on an inverse of the standard error of effect. (**A**) Pain at rest 6 h after surgery; (**B**) Postoperative cumulative 24 h morphine equivalent consumption; (**C**) Pain at rest 24 h after surgery; (**D**) Dynamic pain at 6 h after surgery; (**E**) Dynamic pain at 24 h after surgery; (**F**) The time to first analgesic request.

**Figure 3 jpm-12-00634-f003:**
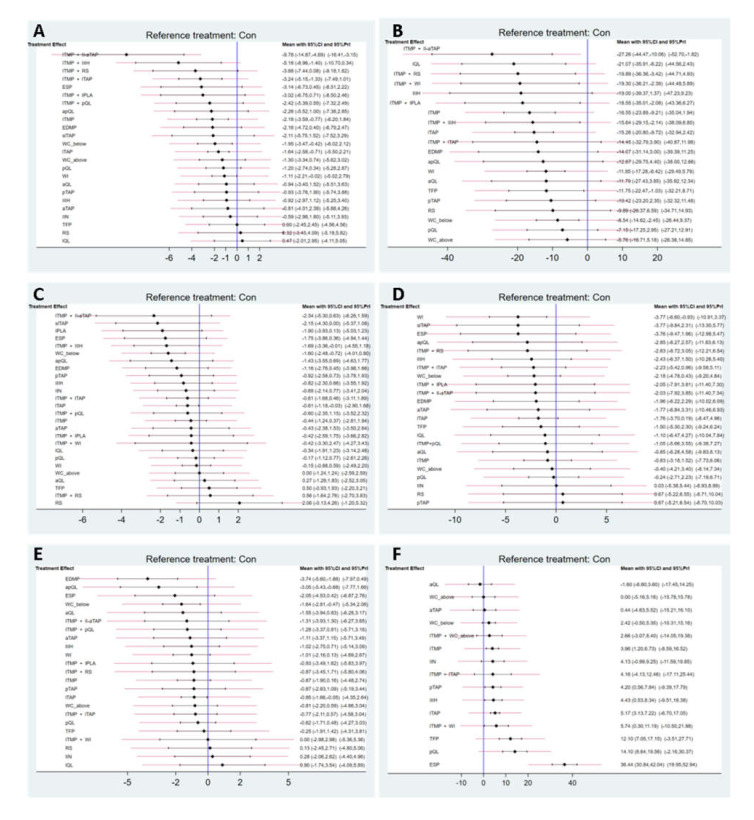
The confidence intervals (CI) and predictive intervals (PrI) of all outcomes. Each solid black line represents the CI for each comparison, and the red one shows the respective PrI. The blue line is the line of no effect (odds ratio = 1). (**A**) Pain at rest 6 h after surgery; (**B**) Postoperative cumulative 24 h morphine equivalent consumption; (**C**) Pain at rest 24 h after surgery; (**D**) Dynamic pain at 6 h after surgery; (**E**) Dynamic pain at 24 h after surgery; (**F**) The time to first analgesic request.

**Figure 4 jpm-12-00634-f004:**
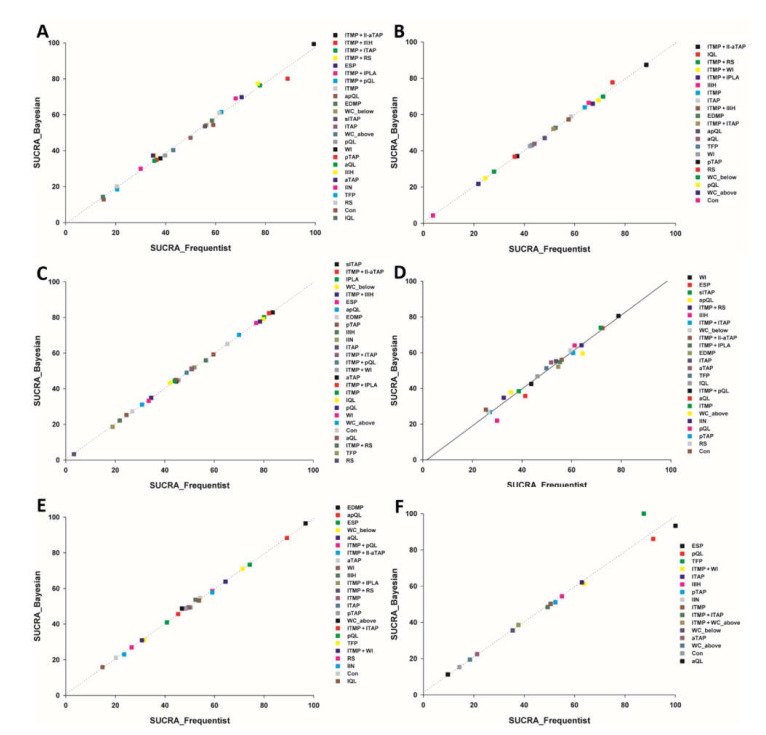
The expected mean ranking and surface under the cumulative ranking curve (SUCRA) values from a frequentist and Bayesian model of all outcomes. SUCRA is a numeric presentation of the overall ranking. The higher the SUCRA value and the closer to 100%, the better the rank of the intervention. (**A**) Pain at rest 6 h after surgery; (**B**) Postoperative cumulative 24 h morphine equivalent consumption; (**C**) Pain at rest 24 h after surgery; (**D**) Dynamic pain at 6 h after surgery; (**E**) Dynamic pain at 24 h after surgery; (**F**) The time to first analgesic request.

**Table 1 jpm-12-00634-t001:** Summary of study characteristics in included studies.

Author/Year	Anesthesia	Groups (*n*)	Management	Supplemental Postoperative Analgesia	Primary Outcome
Aydin et al., 2020 [16]	SA	TFP (30) Control (30)	0.25% bupivacaine 40 mL N/S 40 mL	IV MP PCA (0-1-10); in PACU, fentanyl 50 mcg if pain score > 3;paracetamol 1 g IV q6 h	Postoperative opioid consumption
Aydogmus et al., 2014 [17]	SA	WI (35) slTAP (35)	0.25% levobupivacaine 40 mL 0.25% levobupivacaine 40 mL	If pain score > 3, diclofenac 75 mg IM, then, tramadol 50 mg IV	Pain scores
Baaj et al., 2010 [18]	SA	lTAP (20) Control (20)	0.25% bupivacaine 40 mL N/S 40 mL	IV MP PCA (0-1-10)	MP consumption over 24 h
Bamigboye et al., 2008 [19]	GA	WI (50) Control (50)	0.75% ropivacaine 30 mL N/S 30 mL	Pethidine 100 mg IV q3–4 h; Diclofenac 75 mg IM q12 h; tramadol 37.5 mg and paracetamol 325 mg as needed	Severe pain at 1 h
Barney et al., 2020 [20]	CSE	WC_below (33) Control (38)	0.2% ropivacaine 540 mg and ketorolac 30 mg (5 mL/h) N/S (5 mL/h)	Acetaminophen 975 mg rectal and Ketorolac 15 mg IV in OR; acetaminophen 975 mg q6 h; Ketorolac 15 mg q6 h; Ibuprofen 600 mg q6 h; Oxycodone 5 mg PO for NRS 4–6, 10 mg PO for NRS 7–10.	Pain score with movement at 24 h
Belavy et al., 2009 [21]	SA	lTAP (23) Control (24)	0.5% ropivacaine 40 mL N/S 40 mL	Acetaminophen 1 g rectal; Diclofenac 100 mg after surgery; Acetaminophen 1 g q6 h	MP requirements in 24 h
Bell et al., 2002 [22]	SA	IIIH (31)	0.5% bupivacaine + epinephrine 5 μg/mL 24 mL	IV MP PCA (0–0.02mg/kg−10); Naproxen 500 mg q12 h	24 h IV PCA MP use
		Control (28)	N/S 24 mL		
Bensghir et al., 2008 [23]	SA	WI (20) Control (22)	0.75% ropivacaine 20 mL N/S 20 mL	Paracetamol 1 g q6 h; Ketoprofen 50 mg q6 h; MP 3 mg IV titration if NRS > 3; Tramadol 100 mg IV (max. 400 mg/dL)	No specific comment
Bessmertnyj et al., 2015 [24]	SA	IIIH (54) aTAP (53) Control (57)	0.5% ropivacaine 20 mL 0.25% ropivacaine 40 mL No block	Ketorolac 30 mg IV q8 h; paracetamol 1 g PO q6 h; Tramadol 100 mg IM as needed only in control group	Pain score at rest
Blanco et al., 2015 [25]	SA	pQL (25) Control (23)	0.125% bupivacaine 0.2 mL/kg N/S 0.2 mL/Kg	IV MP PCA (0-1-5); paracetamol 1 g PO q6 h; Diclofenac 50 mg q8 h	MP demands and doses
Blanco et al., 2016 [26]	SA	lQL (38) lTAP (38)	0.125% bupivacaine 0.4 mL/kg 0.125% bupivacaine 0.4 mL/kg	Diclofenac 100 mg rectal and Paracetamol 1 g IV after surgery; IV MP PCA (0-1-5); Paracetamol 1 g PO q6 h; Diclofenac 50 mg q8 h	PCA MP consumption
Bollag et al., 2012 [27]	SA	ITMP + lTAP (25)	SA with MP 100 μg, TAP with 0.375% bupivacaine 40 mL + N/S 1 mL	Ketorolac 30 mg IV during the block; in PACU, MP IV as needed; Acetaminophen 1g q6 h; Diclofenac 75 mg q8 h; Tramadol 50 mg PO q8 h as needed	Wound hyperalgesia 48 h
		ITMP (30)	MP 100 μg, TAP with N/S 41 mL		
Canakci et al., 2018 [28]	SAED	EDMP (40) lTAP (40)	MP 3 mg 0.25% bupivacaine 40 mL	Dex-ketoprofen 50 mg IV as needed	No specific comment
Canovas et al., 2013 [29]	SA	ITMP (30) lTAP (30)	MP 100 μg, TAP with N/S 40 mL 0.5% levobupivacaine 40 mL	IV MP PCA (0-1-10)	No specific comment
Chandon et al., 2014 [30]	SA	lTAP (36) WCI_below (29)	0.375% levobupivacaine 40% 250 mg levobupivacaine in 200 mL (5 mL/h)	Paracetamol 1 g; Ketoprofen 50 mg; Nefopam 20 mg PO q6 h	Pain over 48 h
Corsini et al., 2013 [31]	SA	WI (56) Control (53)	0.5% levobupivacaine 30 mL N/S 30 mL	Paracetamol 1 g IV after surgery; in PACU, MP 3 mg IV if VAS > 4; IV MP PCA (0-1-10); outside OR, Ketoprofen 100 mg q8 h, if VAS > 4	MP consumption
Costello et al., 2009 [32]	SA	ITMP (49)	MP 100 μg, TAP with N/S 40 mL	Ketorolac 30 mg IV; Acetaminophen 1.3 g rectal in OR; in PACU, Diclofenac 50 mg PO q8 h; Acetaminophen 1 g PO q6 h; MP 2 mg IV as needed; then, MP S.Q.; MP 5 mg PO	VAS pain score on movement at 24 h
		ITMP + lTAP (47)	MP 100 μg, 0.375% ropivacaine 40 mL		
Demiraran et al., 2013 [33]	GA	WI (30) Control (30)	0.25% levobupivacaine 20 mL N/S 20 mL	IV PCA with Tramadol (5 mg/h−20 mg−15 min); Diclofenac 75 mg IV if VAS > 3	24 h tramadol consumption
Dereu et al., 2019 [34]	SA	ITMP (82) pTAP (84)	MP 100 μg, TAP with N/S 40 mL 0.5% ropivacaine 40 mL + clonidine 75 μg	Paraceamol 1 g IV and Ketorolac 30 mg IV 1 h after surgery; then, Paracetamol 1 g PO q6 h; Ibuprofen 600 mg PO q8 h	PONV at 24 h
Ducarme et al., 2012 [35]	SA	WI (56) ITMP (44)	0.75% ropivacaine 20 mL MP 100 μg, block with N/S 20 mL	Paraceamol 1 g q6 h; Ketoprofen 50 mg q8 h; Nefopam 20 mg IV; if VAS > 3, MP 3 mg IV	Pain on movement and coughing
Eldaba et al., 2013 [36]	SA	WC_below (40) Control (40)	0.5% bupivacaine 5 mL/h N/S 5 mL/h	IV MP PCA (0-2-10); Ketorolac 30 mg IV q8 h; Acetaminophen 500 mg IV q6 h	No specific comment
Eslamian et al., 2012 [37]	GA	lTAP (24) Control (24)	0.25% bupivacaine 30 mL No block	Tramadol 50 mg IV if needed; Diclofenac 100 mg rectal qd	Pain intensity
Fakor et al., 2014 [38]	SA	lTAP (35) Control (35)	0.25% bupivacaine 40 mL N/S 40 mL	Diclofenac 100 mg rectal if needed	No specific comment
Fusco et al., 2016 [39]	SA	slTAP (48) Control (48)	0.375% levobupivacaine 40 mL N/S 40 mL	In PACU, if VAS > 3, Ketorolac 30 mg IV; then Acetaminophen 1 g IV q6 h if VAS 3–5 or Ketorolac 30 mg IV if VAS 5–7 or Tramadol 100 mg if VAS 7–10	Pain during the first 72 h
Ganta et al., 1994 [40]	GA	IIN (21) WI (20) Control (21)	0.5% bupivacaine 20 mL 0.5% bupivacaine 20 mL No block	No specific comment	No specific comment
Gao et al., 2019 [41]	SA	lTAP Control	0.33% ropivacaine 60 mL No block	In control group, IV PCA with sufentanil	No specific comment
Givens et al., 2002 [42]	ED	WI (20) Control (16)	0.25% bupivacaine 25 mL N/S 25 mL	IV MP PCA (0-1-6)	No specific comment
Hansen et al., 2019 [43]	SA	aQL (34) Control (34)	0.375% ropivacaine 30 mL N/S 30 mL	IV MP PCA (0-5-20); Paracetamol 1 g PO q6 h; Ibuprofen 400 mg q8 h	Opioid consumption
Irwin et al., 2020 [44]	SA	pQL (44) Control (42)	0.25% levobupivacaine 40 mL Sham block	Diclofenac 100 mg and Paracetamol 1 g IV after surgery; IV MP PCA (0-1-5); Paracetamol 1 g PO q6 h; Diclofenac 75 mg PO q12 h	24 h MP consumption
Jadon et al., 2018 [45]	SA	lTAP (67) Control (67)	0.375% ropivacaine 40 mL N/S 40 mL	Declofenac 75 mg IV before the completion of surgery; Diclofenac 75 mg q12 h; Tramadol 50 mg as needed	Time to first analgesic request
Jolly et al.,2 015 [46]	SA	WC_below (34) Control (34)	0.25% levobupivacaine 20 mL bolus, then 1.25 mg/mL, 5 mL/h No procedure	Nefopam 20 mg IV and Acetaminophen 1 g during the surgery; IV MP PCA (0-1.2-7); in PACU, Celecoxib 400 mg; then, Acetaminophen 1 g PO, Nefopam 20 mg PO q6 h	24 h MP consumption
Kagwa et al., 2015 [47]	SA	pTAP (86) Control (84)	0.25% bupivacaine 20–25 mL + 1:400,000 epinephrine Sham block	Paracetamol 1 g and Diclofenac 50 mg PO q8 r	Pain at rest and on movement
Kainu et al., 2012 [48]	CSE	ITMP (24) WC_below (22) Control (20)	MP 160 μg, N/S 5 mL/h WI 0.375% ropivacaine, 5 mL/h N/S 5 mL/h WI	IV oxycodone PCA (0-2-8); after 24 h, Oxycodone 5 or 10 mg PO	24 h oxycodone consumption
Kanazi et al., 2010 [49]	SA	ITMP (28) pTAP (29)	MP 200 μg, TAP with N/S 40 mL 0.375% bupivacaine + epinephrine 5 μg/mL 40 mL	Diclofenac 100 mg rectal q12 h; Paracetamol 1 g IV q6 h; Tramadol 100 mg IV q8 h as needed	Time to first analgesic request
Kang et al., 2019 [50]	SA	pQL (22) aQL (23) apQL (22) EDMP (22)	0.2% ropivacaine 60 mL 0.2% ropivacaine 60 mL 0.2% ropivacaine 60 mL MP 2 mg in 6 mL	Paracetamol 1 g PO q6 h; IV MP PCA (0-0.5-5)	Pain scores at rest and with movement Total MP consumption
Kessous et al., 2012 [51]	GASA	WI (77) Control (76)	1% lidocaine 20 mL N/S 20 mL	Propoxyphene 40 mg and Paracetamol 500 mg for mild pain; meperidine 75 mg for severe pain	No specific comment
Kiran et al., 2017 [52]	SA	lTAP (30) IIIH (30)	0.25% bupivacaine 40 mL 0.25% bupivacaine 40 mL	Paracetamol 1 g IV	Postoperative analgesic sparing
Klasen et al., 2016 [53]	SA	lTAP (25) WC_below (29)	0.75% ropivacaine 3 mL/kg 0.2% ropivacaine 10 mL bolus +5 mL/h	Paracetamol 4 mg PO per day; Ketoprofen 200 mg per day; Nefopam 40 mg per day; IV MP PCA (0-1-10)	48 h MP consumption
Krohg et al., 2018 [54]	SA	lQL (20) control (20)	0.2% ropivacaine 0.4 mg/kg N/S 0.4 mL/kg	Paracetamol 1 g PO; Ibuprofen 400 mg PO q6 h; IV ketobemidone PCA (0-1-8)	24 h ketobemidone consumption
Kwikiriza et al., 2019 [55]	SA	ITMP (65) pTAP (65)	MP 100 μg, Sham block 0.25% bupivacaine + epinephrine 1:200,000 30 mL	Paracetamol 1 g; Diclofenac 50 mg	No specific comment
Lalmand et al., 2017 [56]	SA	ITMP (61) WC_below (63) Control (58)	MP 100 μg, infusion with N/S 0.2% ropivacaine 15 mg bolus + 10 mL/hr Infusion with N/S	Acetaminophen 1 g q6 h; Diclofenac 75 mg q12 h; IV MP PCa (0-1-7)	Time to first analgesic request
Lavand’homme et al., 2007 [57]	SA	WC_above (30) Control (30)	0.2% ropivacaine 240 mL, 5 mL/h N/S 240 mL, 5 mL/h	IV MP PCA (0-1-5); Diclofenac 75 mg IV q12 h; Acetaminophen 1 g q6 h as needed	48 h MP consumption
Lee et al., 2013 [58]	CSE	ITMP + lTAP (25)	MP 250 μg, block with 0.5% ropivacaine 40 mL	In PACU, MP 2 mg IV as needed up to 6 mg; Acetaminophen 1 g PO q6 h for VRS 1–3; Ketorolac 30 mg IV or Ibuprofen 800 mg PO q6 h for VRS 4–5; MP 2 mg IV q10 min as needed or Acetaminophen 600 mg/codeine 60 mg or Oxycodone 10 mg/Acetaminophen 650 mg for VRS 6–10	Pain score with movement at 24 h
		ITMP (24)	MP 250 μg, block with N/S 40 mL		
Loane et al., 2012 [59]	SA	ITMP (33) lTAP (33)	MP 100 μg, sham block 0.5% ropivacaine 3 mg/kg	Naproxen 500 mg recal and Acetaminophen 975 mg IV after surgery; then, Naproxen 500 mg PO q12 h; Acetaminophein 1 g PO q6 h; Hydromorphone 2–4 mg PO q4 h as needed; if still inadequate, PC MP PCA (0-1.5-7)	24 h MP equivalent consumption
Lui et al., 2017 [60]	SA	RS (46)	0.25% bupivacaine 40 mL+ epinephrine 5 μg/mL	Paracetamol 1 g; Tramadol 50 mg	Pain on movement
		ITMP + RS (47) ITMP (38)	MP 100 μg, block with same regimen MP 100 μg, block with N/S 40 mL		
Magnani et al., 2006 [61]	SA	ITMP + WC_ above (10)	MP 50 μg, infusion with 0.2% levobupivacaine 2 mL/h	No specific comment	No specific comment
		ITMP (10)	MP 50 μg		
Malawat et al., 2020 [62]	SA	ESP (30) lTAP (30)	0.2% ropivacaine 0.2 mL/kg 0.2% ropivacaine 0.2 mL/kg	Diclofenac 75 mg	Time to first analgesic request
Mankikar et al., 2016 [63]	SA	pTAP (30) Control (30)	0.5% ropivacaine 30 mL N/S 30 mL	Paracetamol 1 g IV after surgery	No specific comment
McKeen et al., 2014 [64]	SA	ITMP + lTAP (35)	MP 100 μg, block with 0.25% ropivacaine 40 mL	Ketorolac 30 mg IV and Acetaminophen 1 g IV before block; Naproxen 250 mg q8 h; Acetaminophen 1 g q6 h; Oxycodone 2.5–5 mg q6 h as needed	Pain score, Quality of recovery, 24 h opioid consumption
		ITMP (39)	MP 100 μg, block with N/S 40 mL		
McMorrow et al., 2011 [65]	SA	ITMP (20)	MP 100 μg, block with N/S	Paracetamol 1 g and Diclofenac 100 mg after surgery; Paracetamol 1 g PO q6 h; Diclofenac 100 mg rectal at 18 h; IV MP PCA (0-1-5)	Pain on movement
		ITMP + lTAP (20)	MP 100 μg, block with 0.375% bupivacaine 2 mg/kg		
		lTAP (20) Control (20)	0.375% bupivacaine 2 mg/kg Block with N/S		
Mecklem et al.,1995 [66]	SA	RS (35) Control (35)	0.25% bupivacaine 20 mL #8 N/S 20 mL #8	IV MP PCA (0-1-5)	No specific comment
Mieszkowski et al., 2018 [67]	SA	lQL (30) Control (28)	0.375% ropivacaine 48 mL No block	Paracetamol 1 g IV before block; Paracetamol 1 g IV q6 h; if NRS > 3, MP 5 mg S.C.	48 h MP consumption
Naghshineh et al., 2015 [68]	GA	IIN (40) Control (40)	0.5% bupivacaine 20 mL No block	Pethidine bolus	No specific comment
Niklasson et al., 2012 [69]	SA	WI (130)	0.25% bupivacaine + epinephrine 5 μg/mL 40 mL	Paracetamol 1 g q6h; MP IV if needed; after 24 h, Codeine 75 mg PO q6 h; Ibuprofen 200 mg q6 h	12 and 24 h MP consumption
		Control (130)	N/S 40 mL		
O’Neill et al., 2012 [70]	SA CSE	WC_below (29) ED MP (29)	1% ropivacaine 10 mL bolus, 5 mL/h MP 2 mg/mL q12 h #4	Acetaminophen 1 g IV q6 h; Diclofenac 75 mg IM as needed	Pain score at rest at 24 h
Patel et al., 2017 [71]	SA	ITMP + IPLA (99)	MP 100 μg, 2% lidocaine 20 mL + 1:200,000 epinephrine	Ketorolac 30 mg IV and Acetaminophen 1.3 g suppository after surgery; in PACU MP 2 mg IV as needed; then, Diclofenac 50 mg PO q8 h; Acetaminophen 1 g q6 h; MP 2 mg S.C./IV or Hydromorphone 0.4 mg as needed	Pain score on movement at 24 h
		ITMP (94)	MP 100 μg, N/S 20 mL		
Rackelboom et al., 2010 [72]	SA	ITMP + WC_ above (25) ITMP + WC_ below (25)	MP 100 μg, Ropivacaine 450 mg + Ketoprofen 200 mg + N/S 240 mL, 5 mL/h in both group	IV MP PCA (0-1.5-7)	48 h MP consumption
Reinikainen et al., 2014 [73]	SA	WC_above (33) Control (34)	0.75% ropivacaine 100 mL, 2 mL/h N/S 100 mL, 2 mL/h	Paracetamol 1 g q8 h; Ibuprofen 600 mg q8 h PO; Oxycodone 0.2 mg/kg IM (NRS > 3) or 0.05 mg/kg IV (NRS > 7)	48 h oxycodone consumption
Salama et al., 2020 [74]	SA	ITMP (30) pQL (30) control (30)	MP 100 μg, block with N/S 0.375% ropivacaine 48 mL Block with N/S 48 mL	Paracetamol 1 g IV and Diclofenac 100 mg suppository after surgery; IV MP PCA (0-1-5); if NRS > 3, Paracetamol 1 g IV	Pain score at rest and on movement
Sekhavat et al., 2011 [75]	GA	WI (52) Control (52)	2% lidocaine 10 mL N/S 10 mL	Mefenamic acid 500 mg PO q4 h; MP 5 mg IM as needed	No specific comment
Serifsoy et al., 2020 [76]	GA	TFP (35) Control (35)	0.5% bupivacaine 20 mL + 2% lidocaine 10 mL + N/S 20 mL No block	IV Tramadol PCA (0-10-20); in PACU, NRS > 4 Fentanyl 25 µg; then, Paracetamol 1 g IV q8 h; Diclofenac 75 mg IM (NRS > 4)	24 h tramadol consumption
Shahin et al., 2010 [77]	SA	IPLA (176) Control (178)	2% lidocaine 10 mL N/S 10 mL	Acetaminophen 1 g q6 h; Ibuprofen 10 mg suppository; Ibuprofen 500 mg PO q4–6 h; MP 2 mg IV as needed	Epigastric pain on 1st and 5th day
Singh et al., 2013 [78]	SA	ITMP + lTAP (20)	MP 150 μg, block with 0.5% ropivacaine 3 mg/kg in 60 mL	Ketorolac 30 mg IV during surgery; Ketorolac 30 mg IV; Acetaminophen 650 mg PO q6 h; Codeine 30 mg PO or Oxycodone 5–10 mg q4 h as needed	Pain score difference on movement at 24 h
		ITMP (20)	MP 150 μg, block with N/S 60 mL		
Srivastava et al., 2015 [79]	SA	lTAP (31) control (31)	0.25% bupivacaine 40 mL Sham block	Diclofenac 75 mg IV q8 h; IV Tramadol PCA (0-20-10)	Additional analgesics during 48 h
Staker et al., 2018 [80]	SA	ITMP + II-aTAP (50)	MP 150 μg, 0.33% ropivacaine 50 mL (200 mg)	Paracetamol 1.5 g suppository and Diclofenac 100 mg after surgery; in PACU, Fentanyl 10 μg (NRS < 7), 20 μg (NRS > 7); IV Fentanyl PCA (0–10 μg−5); Paracetamol 1 g PO q6 h; Diclofenac 50 mg PO q8 h	Difference in fentanyl dose at 24 h
		ITMP (50)	MP 150 μg, Sham block		
Svirskiǐ et al., 2012 [81]	SA	lTAP (31) control (31)	0.375% ropivacaine 40 mL No block	In TAP group: Ketoprofen 100 mg IV q12 h; Paracetamol 1 g IV q8 h In control group: Paracetamol 1 g q6 h; Ketoprofen 100 mg every 8–12 h; Tramadol 100 mg as needed	No specific comment
Pavy et al., 1994 [82]	SA	ITMP + WI (20)	MP 250–300 μg, infiltration with 0.5% bupivacaine 20–30 mL	Codeine 30 mg PO; Paracetamol 325 mg q3 h	No specific comment
		ITMP (20)	MP 250–300 μg, N/S 20–30 mL		
Tamura et al., 2019 [83]	SA	ITMP + pQL (34)	MP 100 μg, block with 0.75% ropivacaine 0.9 mL/kg	Droperidol 1.25 mg and Fentanyl 90 μg and Acetaminophen 15 mg/kg IV after baby out; Pentazocine 15 mg IV (NRS 3–6), Pentazocine 15 mg and Acetaminophen 15 mg/kg IV (NRS > 6)	Pain score at 6 h
		ITMP (38) pQL (36) control (38)	MP 100 μg, block with N/S 0.75% ropivacaine 0.9 mL/kg N/S 0.9 mL/kg		
Tan et al., 2012 [84]	GA	lTAP (20) control (20)	0.25% levobupviacaine 40 mL No block	IV MP PCA (0-1-5)	24 h MP consumption
Tawfik et al., 2017 [85]	SA	WI (39) lTAP (39)	0.25% bupivacaine 30 mL 0.25% bupivacaine 40 mL	Ketorolac 30 mg IV q8 h; Paracetamol 1 g PO q8 h; IV Fentanyl PCA (0–20 μg−7)	24 h fentanyl consumption
Telnes et al., 2015 [86]	SA	lTAP (28)	0.25% bupivacaine 40 mL + epinephrine 5 μg/mL	Paracetamol 1 g PO q6 h; Diclofenac 50 mg PO q8 h; IV MP PCA (0-1-6)	48 h MP consumption
		WI (29)	0.25% bupivacaine 20 mL + epinephrine 5 μg/mL		
Triyasunant et al., 2015 [87]	SA	ITMP + WI (28)	MP 200 μg, infiltration with 0.125% bupivacaine 40 mL	Parecoxib 40 mg IV (after surgery, 12 h); IV MP PCA (0-1-5)	Pain free period
		ITMP (28)	MP 200 μg, no block		
Trotter et al., 1991 [88]	GA	WI (14) Control (14)	0.5% bupivacaine 20 mL N/S 20 mL	IV MP PCA (0-2-10)	No specific comment
Vallejo et al., 2012 [89]	SA	ITMP + IIIH (17)	MP 150–200 μg, 0.5% bupivacaine 20 mL	Ketorolac 30 mg IV q6 h for 24 h; then, Ibuprofen, Oxycodone, Acetaminophen/Oxycodone, Acetaminophen/Hydrocodone PO	Pain score at 48 h
		ITMP (17)	MP 150–200 μg, block with N/S 20 mL		
Wagner-Kovacec et al., 2018 [90]	SA	WC_above (15) Control (15)	0.25% levobupivacaine 270 mL, 5 mL/h N/S 270 mL, 5 mL/h	Paracetamol 1 g IV q6 h; Piritramide 2 mg IV as needed	24 and 48 h piritramide consumption
Wolfson et al., 2012 [91]	SA	ITMP + IIIH (17)	MP 200 μg, block with 0.5% bupivacaine 24 mL	Ketorolac 30 mg IV; Acetaminophen 1 mg/ Oxycodone 10 mg PO q6 h; IV MP PCA (0-2-10)	Pain score at rest at 24 h
		ITMP (17)	MP 200 μg, N/S 24 mL		

IV MP PCA setting was presented in the following order: basal infusion-bolus dose (mg)-lockout time. N/S: normal saline; IV: intravenous; PCA: patient controlled analgesia; MP: morphine; IM: intramuscular; PO: per os; NRS: numerical rating scale; EDMP: epidural morphine; WCI: wound closure infiltration; VAS: visual analogue scale; PONV: postoperative nausea and vomiting; qd: per day; S.C.: subcutaneous; ITMP: intrathecal morphine; ESP: erector spinae plane block; pQL: posterior quadratus lumborum block; lQL: lateral quadratus lumborum block; aQL: anterior quadratus lumborum block; apQL: combined anterior and posterior quadratus lumborum block; lTAP: lateral transversus abdominis plane block; aTAP: anterior transversus abdominis plane block k; pTAP: posterior transversus abdominis plane block; TFP: transverse fascial plane block; TAP: transversus abdominis plane block; IIIH: ilioinguinal-iliohypogastric nerve block; IIN: ilioinguinal nerve block; WI: wound infiltration; WC: wound continuous infusion; IPLA: intraperitoneal local anesthetics; GA: general anesthesia; SA: spinal anesthesia; ED: epidural anesthesia; CSE: combined spinal epidural anesthesia; PCA: patient controlled analgesia; N/S: 0.9% normal saline; PACU: post-anesthetic care unit; OR: operating room; VRS: verbal rating scale.

**Table 2 jpm-12-00634-t002:** The GRADE evidence quality for each outcome.

Outcomes	Number of Studies/Patients	Quality Assessment								Quality
		Downgrade				Upgrade				
		Study Limitation	Inconsistency	Indirectness	Imprecision	Publication Bias	Large Effect	Dose-Response	Confounding	
Pain at rest 6 h after surgery	59/4622	serious	not serious	not serious	not serious	not serious	no	no	no	⨁⨁⨁◯ Moderate
Postoperative cumulative 24 h morphine equivalent consumption	44/3360	serious	not serious	not serious	not serious	not serious	no	no	no	⨁⨁⨁◯ Moderate
Pain at rest 24 h after surgery	59/4697	serious	serious	not serious	not serious	not serious	no	no	no	⨁⨁◯◯ Low
Dynamic pain at 6 h after surgery	37/2837	serious	serious	not serious	not serious	not serious	no	no	no	⨁⨁◯◯ Low
Dynamic pain at 24 h after surgery	44/3371	serious	serious	not serious	not serious	not serious	no	no	no	⨁⨁◯◯ Low
The time to first analgesic request	24/1812	serious	not serious	not serious	not serious	not serious	no	no	no	⨁⨁⨁◯ Moderate

GRADE: Grading of recommendations assessment, development, and evaluation system; ⨁⨁⨁◯: moderate quality; ⨁⨁◯◯; low quality.

## Data Availability

The datasets used and analyzed during the current study are available from the corresponding author upon reasonable request.

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
