# Peer review of "Postoperative Analgesic Effectiveness of Peripheral Nerve Blocks in Cesarean Delivery: A Systematic Review and Network Meta-Analysis"

_jpm, 2022, doi:10.3390/jpm12040634_

Round 1
Reviewer 1 Report
Dear Author
I read this article as titled "Postoperative analgesic effectiveness of peripheral nerve blocks in cesarean delivery".
The design of the study is well understood and the description was detail oriented.
The author has been able to discuss this challenge of caring for these patients.
The subject was also somehow new and more practices need to be done.
In my opinion the study was almost perfect.
Sincerely
Author Response
Dear Author
I read this article as titled "Postoperative analgesic effectiveness of peripheral nerve blocks in cesarean delivery".The design of the study is well understood and the description was detail oriented.
The author has been able to discuss this challenge of caring for these patients.
The subject was also somehow new and more practices need to be done.
In my opinion the study was almost perfect.
Sincerely
Our response: We sincerely appreciate to reviewer’s good comment on our work. Your support will be greatly helpful to our future study. Thank you once more.
Reviewer 2 Report
Hi,
The literature on this subject is very well summarized.
The information in figure 3 and figure 4 is not read.
The analysis of the studies is very difficult, but they are analyzed very well.
Good luck with your work.
Author Response
Hi,
The literature on this subject is very well summarized.
The information in figure 3 and figure 4 is not read.
The analysis of the studies is very difficult, but they are analyzed very well.
Good luck with your work.
Our response: We sincerely appreciate to reviewer’s good comment on our work. Your support will be greatly helpful to our future study. Thank you once more.
Reviewer 3 Report
Dear Authors,
I read your work with great interest and I'd like to congratulate with you. It is a very well written paper, with an accurate design and of major interest.
I believe it should undergo a minor spelling and grammar revision, but in my opinion it is a work of great value.
Author Response
Dear Authors,
I read your work with great interest and I'd like to congratulate with you. It is a very well written paper, with an accurate design and of major interest.
I believe it should undergo a minor spelling and grammar revision, but in my opinion it is a work of great value.
Our response: We sincerely appreciate to reviewer’s good comment on our work. Your support will be greatly helpful to our future study. I got a native English editing service. Thank you once more.